# Attacking Graph Neural Networks with Bit Flips: Weisfeiler and Lehman Go Indifferent

## Abstract

Prior attacks on graph neural networks have mostly focused on graph poisoning and evasion, neglecting the network's weights and biases. Traditional weight-based fault injection attacks, such as bit flip attacks used for convolutional neural networks, do not consider the unique properties of graph neural networks. We propose the Injectivity Bit Flip Attack, the first bit flip attack designed specifically for graph neural networks. Our attack targets the learnable neighborhood aggregation functions in quantized message passing neural networks, degrading their ability to distinguish graph structures and losing the expressivity of the Weisfeiler-Lehman test. Our findings suggest that exploiting mathematical properties specific to certain graph neural network architectures can significantly increase their vulnerability to bit flip attacks. Injectivity Bit Flip Attacks can degrade the maximal expressive Graph Isomorphism Networks trained on various graph property prediction datasets to random output by flipping only a small fraction of the network's bits, demonstrating its higher destructive power compared to a bit flip attack transferred from convolutional neural networks. Our attack is transparent and motivated by theoretical insights which are confirmed by extensive empirical results.

## 1 Introduction

Graph neural networks (GNNs) are a powerful machine learning technique for handling structured data represented as graphs with nodes and edges. These methods are highly versatile, extending the applicability of deep learning to new domains such as financial and social network analysis, medical data analysis, and chem- and bioinformatics (Lu & Uddin, 2021; Cheung & Moura, 2020; Sun et al., 2021; Gao et al., 2022b; Wu et al., 2018; Xiong et al., 2021). With the increasing adoption of GNNs, there is a pressing need to investigate their potential security vulnerabilities. Traditional adversarial attacks on GNNs have focused on manipulating input graph data (Wu et al., 2022) through poisoning attacks, which result in the learning of a faulty model (Ma et al., 2020; Wu et al., 2022), or evasion attacks, which use adversarial examples to degrade inference. These attacks can be targeted (Zügner et al., 2018) or untargeted (Zügner & Günnemann, 2019; Ma et al., 2020) and involve modifications to node features, edges, or the injection of new nodes (Sun et al., 2020; Wu et al., 2022). Targeted attacks degrade the model's performance on a subset of instances, while untargeted attacks degrade the model's overall performance (Zhang et al., 2022a). A classification of existing graph poisoning and evasion attacks and defense mechanisms as well as a repository with representative algorithms can be found in the comprehensive reviews by Jin et al. (2021) and Dai et al. (2022).

Prior research on the security of GNNs has not yet considered the potential of Bit Flip Attacks (BFAs) (Jin et al., 2021; Xu et al., 2020; Wu et al., 2022; Ma et al., 2020), which directly manipulate a target model at inference time, and previous works on BFA have not yet explored attacks on GNNs (Qian et al., 2023). While the general methods can be transferred to GNNs, they do not consider their unique mathematical properties, although it has been observed that adapting BFAs to the specific properties of a target network can increase harm (Venceslai et al., 2020) and that BFAs can be far from optimal on non-convolutional models (Hector et al., 2022). We address this research gap by exploring the effects of malicious perturbations on a GNN's trainable parameters and their impact on prediction quality. Specifically, we target the expressivity of GNNs, i.e., their ability to distinguish non-isomorphic graphs or node neighborhoods. The most expressive GNNs based on message passing, including the widely-used Graph Isomorphism Networks (GINs) (Xu et al., 2019), have the same discriminative power as the 1-Weisfeiler-Lehman test (1-WL) for suitably

parameterized neighborhood aggregation functions (Morris et al., 2019; Xu et al., 2019; Morris et al., 2021c). Our attack targets these parameters to degrade the network's ability to differentiate between non-isomorphic structures.

Previous research has shown that convolutional neural networks (CNNs) prominent in the computer vision domain are highly susceptible to BFAs (Hong et al., 2019). These works typically focus on CNNs to which techniques like pruning or quantization are applied to improve efficiency, e.g., (Rakin et al., 2019; Lee & Chandrakasan, 2022). Efficient implementation is likewise crucial for practical applications of GNNs, making it necessary to investigate the interaction between robustness and efficiency. Recently, efforts have been made to develop quantization methods for GNNs (Bahri et al., 2021; Feng et al., 2020) and technical realizations for their deployment (Shyam et al., 2021; Zhu et al., 2023) as well as as potential applications exist, e.g., (Dong et al., 2023; Derrow-Pinion et al., 2021; Bertalanič & Fortuna, 2023). However, the relationship between these techniques and robustness has not yet been studied. The only work on GNNs resilience to bit flips studies random bit faults in floating-point hardware (Jiao et al., 2022).

**Related work** The security issue of BFAs is recognized for quantized CNNs, which are used in critical applications like medical image segmentation (Zhang & Chung, 2021; Askari-Hemmat et al., 2019) and diagnoses (Ribeiro et al., 2022; Garifulla et al., 2021). In contrast, GNNs used in safety-critical domains like medical diagnoses (Li et al., 2020b; Gao et al., 2022b; Lu & Uddin, 2021), electronic health record modeling (Liu et al., 2020c; Sun et al., 2021), and drug development (Lin et al., 2020; Xiong et al., 2021; Cheung & Moura, 2020) have not been sufficiently studied for their robustness against BFAs. Qian et al. (2023) and Khare et al. (2022) distinguish between targeted and untargeted BFAs similar to the distinction made between targeted and untargeted poisoning and evasion techniques. The high volume of related work on BFAs for quantized CNNs (Rakin et al., 2022; Roohi & Angizi, 2022; Khare et al., 2022; Qian et al., 2023; Rakin et al., 2020; Chen et al., 2021; Ghavami et al., 2022a;b; Park et al., 2021; Breier et al., 2022; Venceslai et al., 2020) based on the seminal work by Rakin et al. (2019) and associated BFA defense mechanisms (He et al., 2020; Li et al., 2020a; Javaheripi & Koushanfar, 2021; Li et al., 2021; Guo et al., 2021; Javaheripi et al., 2022; Liu et al., 2020b; 2023; Zhang et al., 2022b; Khoshavi et al., 2021; Hector et al., 2022; Özdenizci & Legenstein, 2022; Poduval et al., 2022) published recently underscores the need for research in the direction of both BFA and defense mechanisms for quantized GNNs. We represent these traditional BFA methods by Progressive Bitflip Attack (Rakin et al., 2019) as most other BFA variants are based on it. We will subsequently refer to this specific BFA as **PBFA** and use the term BFA for the broader class of bit flip attacks only.

**Contribution** We investigate the susceptibility of GNNs to PBFA in a motivating case study, which reveals that PBFA fails to significantly outperform random bit flips on tasks requiring the structural discrimination of graphs. To overcome the problem, we introduce the Injectivity Bit Flip Attack (**IBFA**), a novel attack targeting the discriminative power of neighborhood aggregation functions used by GNNs. Specifically, we investigate the maximal expressive architecture GIN, where this function is injective for suitable parameters (Xu et al., 2019), and which is integrated in popular frameworks (Fey & Lenssen, 2019) and widely used in practice, e.g., (Gao et al., 2022c; Wang et al., 2023b; Yang et al., 2022; Peng et al., 2020; Bertalanič & Fortuna, 2023; Liu & Wang, 2021). IBFA differs from existing BFA variants for CNNs by its bit-search algorithm's optimization goal as well as input data selection strategy and is distinct from graph poisoning and evasion attacks as input data is left modified. We provide a strong theoretical fundament for IBFA which is confirmed by extensive experimental evidence of its effectiveness on real-world datasets under the assumptions common in established BFA research, which we discuss in sec. 4. Our results indicate that IBFA dominates its baselines in terms of both degradation of prediction quality and the number of bit flips required to degrade GIN's output to random, effectively rendering it indifferent to graph structures.

## 2 PRELIMINARIES

IBFA targets quantized neighborhood aggregation based GNNs by exploiting their expressivity, which can be theoretically linked to the 1-Weisfeiler-Lehman (1-WL) graph isomorphism test. Hence, we initiate our discourse by providing a concise overview of such GNNs, the 1-WL algorithm, quantization, PBFA and introduce our notation and terminology along the way.

**Graph theory** A *graph G* is a pair $(V, E)$ of a finite set of *nodes V* and a finite set of *edges* $E \subseteq \{\{u, v\} \subseteq V\}$. The set of nodes and edges of $G$ is denoted by $V(G)$ and $E(G)$, respectively. The *neighborhood* of $v$ in $V(G)$ is $N(v) = \{u \in V(G) \mid \{v, u\} \in E(G)\}$. If there exists a bijection $\varphi \colon V(G) \to V(H)$ with $\{u, v\}$ in $E(G)$ if and only if $\{\varphi(u), \varphi(v)\}$ in $E(H)$ for all $u, v$ in $V(G)$, we call the two graphs $G$ and $H$ *isomorphic* and write $G \simeq H$. For two graphs with roots $r \in V(G)$ and $r' \in V(H)$, the bijection must additionally satisfy $\varphi(r) = r'$. The equivalence classes induced by $\simeq$ are referred to as *isomorphism types*.

A function $V(G) \to \Sigma$ with arbitrary codomain $\Sigma$ is called a *node coloring*. Then, a *node colored* or *labeled graph* $(G, l)$ is a graph $G$ endowed with a node coloring $l$. We call $l(v)$ a *label* or *color* of $v \in V(G)$. A node coloring $c$ *refines* a node coloring $d$, denoted $c \sqsubseteq d$ if $\forall v, w \in V(G) \colon c(v) = c(w) \Rightarrow d(v) = d(w)$. If $c \sqsubseteq d$ and $d \sqsubseteq c$, then two colorings are *equivalent*, which is denoted by $c \equiv d$. The maximal subset $Q \subseteq V(G)$ with $c(v) = c(w)$ for all $v, w \in Q$ is called a *color class* of a node coloring $c$. Further, let $\Pi$ be a partition of $G$. We refer to $\Pi$ as *equitable* if $\forall P, Q \in \Pi$ it holds that $\forall u, v \in P \colon |N(u) \cap Q| = |N(v) \cap Q|$ (Kiefer, 2020, p. 19). Every graph has a unique coarsest equitable partition (Lerner, 2004, p. 239) which is precisely the partition 1-WL's colors (see sec. 2) induce on the node set after termination (Kiefer, 2020, p. 30). We denote a multiset by $\{\!\{\ldots\}\!\}$.

**The Weisfeiler-Lehman algorithm** We outline the methodology of the 1-WL algorithm for labelled graphs. Let $(G, l)$ denote a labelled graph. In every iteration $t > 0$, a node coloring $c_l^{(t)} \colon V(G) \to \Sigma$ is computed, which depends on the coloring $c_l^{(t-1)}$ of the previous iteration. At the beginning, the coloring is initialized as $c_l^{(0)} = l$. In subsequent iterations $t > 0$, the coloring is updated according to

$$c_l^{(t)}(v) = \text{HASH}\left(c_l^{(t-1)}(v), \{\!\{c_l^{(t-1)}(u) | u \in N(v)\}\!\}\right), \tag{1}$$

where HASH is an injective mapping of the above pair to a unique value in $\Sigma$, that has not been used in previous iterations. The HASH function can, for example, be realized by assigning new consecutive integer values to pairs when they occur for the first time (Shervashidze et al., 2011). Let $C_l^{(t)}(G) = \{\!\{c_l^{(t)}(v) \mid v \in V(G)\}\!\}$ be the multiset of colors a graph exhibits in iteration $t$. The iterative coloring terminates if $|C_l^{(t-1)}(G)| = |C_l^{(t)}(G)|$, i.e., the number of colors does not change between two iterations. For testing whether two graphs $G$ and $H$ are isomorphic, the above algorithm is run in parallel on both $G$ and $H$. If $C_l^{(t)}(G) \neq C_l^{(t)}(H)$ for any $t \geq 0$, then $G$ and $H$ are not isomorphic. Non-isomorphic graphs $G$ and $H$ with $C_l^{(t)}(G) = C_l^{(t)}(H)$ for all $t \in \mathbb{N}$ exist, but they are rare even for $t = 2$ (Babai & Kucera, 1979) and many real-world graph learning benchmarks do not contain such graph pairs (Morris et al., 2021b; Zopf, 2022). The label assigned to a node $v$ in the $t$th iteration of the 1-WL test can be understood as a tree representation of the $t$-hop neighborhood of $v$, in the sense that there is a bijection between labels in $\Sigma$ and the isomorphism types of such trees of height $t$, see def. 1 and (D'Inverno et al., 2021; Jegelka, 2022; Schulz et al., 2022) for details.

In social network analysis, structural roles refer to groups of nodes that share similar local structural characteristics. On the other hand, communities refer to sets of nodes that have a higher number of connections within the group compared to those outside of it. While structural roles and communities are distinct concepts, they are both significant and complementary in nature (Lerner, 2004; Rossi et al., 2020). Structural roles can be linked to 1-WL via the equitable partition given by 1-WL's color classes (Lerner, 2004, p.p. 239).

**Graph neural networks** GNNs utilize graph structure and node features to derive a representation vector for a specific node, denoted as $\mathbf{h}_v$, or for the entire graph, denoted as $\mathbf{h}_G$. Contemporary GNNs employ a neighborhood aggregation or message passing approach, in which the representation of a node is iteratively updated through the aggregation of representations of its neighboring nodes. Upon completion of $k$ iterations of aggregation, the representation of a node encapsulates the structural information within its $k$-hop neighborhood (Xu et al., 2019). The $k$th layer of a GNN computes the node features $\mathbf{h}_v^{(k)}$ formally defined by

$$\mathbf{a}_v^{(k)} = \text{AGGREGATE}^{(k)}\left(\{\!\{\mathbf{h}_u^{(k-1)} \mid u \in N(v)\}\!\}\right), \quad \mathbf{h}_v^{(k)} = \text{COMBINE}^{(k)}\left(\mathbf{h}_v^{(k-1)}, \mathbf{a}_v^{(k)}\right). \tag{2}$$

Initially, $\mathbf{h}_v^{(0)}$ are the node features of the given graph. The choice of AGGREGATE$^{(k)}$ and COMBINE$^{(k)}$ in GNNs is critical, and several variants have been proposed (Xu et al., 2019).

**Graph isomorphism network** GIN is characterized by its simplicity and has been proven to possess the highest level of expressivity among GNNs based on neighborhood aggregation. It has the same discriminative power as the 1-WL test in distinguishing non-isomorphic graphs (Xu et al., 2019). A large body of work is devoted to GNNs exceeding this expressivity (Morris et al., 2021b;c). However, neighborhood aggregation is widely used in practice and 1-WL is sufficient to distinguish most graphs in common benchmark datasets (Morris et al., 2021b; Zopf, 2022). As established by (Xu et al., 2019), a neighborhood aggregation GNN with a sufficient number of layers can reach the same discriminative power as the 1-WL test if both the AGGREGATE and COMBINE functions in each layer's update rule as well as its graph level READOUT are injective. GIN achieves this with the update rule

$$\mathbf{h}_v^{(k)} = \text{MLP}^{(k)} \left( (1 + \epsilon^{(k)}) \cdot \mathbf{h}_v^{(k-1)} + \sum_{u \in N(v)} \mathbf{h}_u^{(k-1)} \right) \tag{3}$$

integrating a multi layer perceptron (MLP) into COMBINE$^{(k)}$, realizing a universal function approximator on multisets (Hornik et al., 1989; Hornik, 1991; Zaheer et al., 2017). If input features are one-hot encodings, an MLP is not needed before summation in the first layer, since summation is injective in this case. For graph level readout, GIN employs concatenation of the sums of all node features of the same layer according

$$\mathbf{h}_G = \Big\|_{k=0}^{n} \text{READOUT} \left( \{\!\{\mathbf{h}_v^{(k)} \mid v \in G\}\!\} \right), \tag{4}$$

where $\|$ denotes the concatenation of vectors. The approach provably generalizes the WL test and the WL subtree kernel (Xu et al., 2019).

**Quantization** Quantization involves either a reduction in the precision of the weights, biases, and activations within a neural network or the use of a more efficient representation, resulting in a decrease in model size and memory utilization (Kummer et al., 2023). Our proposed work explores how maliciously induced bit flips in quantized weights and biases degrade the model's quality metrics. In accordance with the typical set-up chosen in the related work on BFA, see sec. 1, we apply scale quantization to map `FLOAT32` tensors to the `INT8` range, and

$$\mathcal{Q}(\mathbf{W}_l) = \mathbf{W}_{q,l} = \text{clip}(\lfloor \mathbf{W}_l/s \rceil, a, b), \qquad \mathcal{Q}^{-1}(\mathbf{W}_{q,l}) = \widehat{\mathbf{W}}_l = \mathbf{W}_{q,l} \times s \tag{5}$$

specifies such a quantization function $\mathcal{Q}$ and its associated dequantization function $\mathcal{Q}^{-1}$. In eq. 5, $s$ is the scaling parameter, $\text{clip}(x, a, b) = \min(\max(x, a), b)$ with $a$ and $b$ the maximum and minimum thresholds (also known as the quantization range), $\lfloor \ldots \rceil$ denotes nearest integer rounding, $\mathbf{W}_l$ is the weight of a layer $l$ to be quantized, $\mathbf{W}_{q,l}$ its quantized counterpart and $\widehat{\ldots}$ indicates a perturbation (i.e., rounding errors in the case of eq. 5). Similar to other works on BFA that require quantized target networks, e.g., (Rakin et al., 2019), we address the issue of non-differentiable rounding and clipping functions in $\mathcal{Q}$ by employing Straight Through Estimation (STE) (Bengio et al., 2013).

**Progressive bit flip attack** PBFA, introduced in seminal work by Rakin et al. (2019), utilizes a quantized trained CNN $\Phi$ and employs the progressive bit search (**PBS**) to identify which bit flips will damage the CNN most. PBS begins with a single forward and backward pass, conducting error backpropagation without weight updates on a randomly selected batch $\mathbf{X}$ of training data with a target vector $\mathbf{t}$. It then selects the weights linked to the top-$k$ largest binary encoded gradients as potential candidates for flipping the associated bits. These candidate bits are iteratively tested (flipped) across all $L$ layers to find the bit that maximizes the difference between the loss $\mathcal{L}$ of the perturbed and the loss of the unperturbed CNN, whereby the same loss function is used that was minimized during training, e.g., (binary) cross entropy (CE) for (binary) classification.

$$\max_{\{\widehat{\mathbf{W}}_{q,l}\}} \quad \mathcal{L}\Big(\Phi(\mathbf{X}; \{\widehat{\mathbf{W}}_{q,l}\}_{l=1}^{L}), \mathbf{t}\Big) - \mathcal{L}\Big(\Phi(\mathbf{X}; \{\mathbf{W}_{q,l}\}_{l=1}^{L}), \mathbf{t}\Big) \tag{6}$$

The source of the perturbation $\widehat{\ldots}$ in eq. 6 are adversarial bit flips. If a single bit flip does not improve the optimization goal, PBS is executed again, considering combinations of 2 or more bit flips. This process continues until the desired level of network degradation is achieved or a predefined number of iterations is reached. Further details on PBFA/PBS can be found in the appendix.

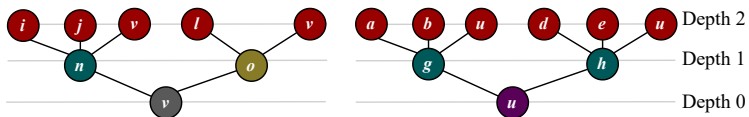

Figure 1: Example of two non-isomorphic unfolding trees $T^{(2)}(u) \not\simeq T^{(2)}(v)$ of height 2 associated with the nodes $u$ and $v$. A function solving a WL-discrimination task for $k = 2$ must be able to discriminate $u$ and $v$ based on the structure of their unfolding trees.

## 3 THEORETICAL FRAMEWORK

In principle, PBFA as described above and potentially most other BFA variants based on it can be directly ported to GNNs. However, Hector et al. (2022) have demonstrated that the high susceptibility of CNNs to BFA is closely tied to weight sharing in convolutional filters. The absence of such filters in GNNs motivates the development of a specialized attack for GNNs.

In our preliminary case study, which in full is contained in the appendix, we examine PBFA's effectiveness on various GNN architectures. The case study's results indicate that quantized GINs trained on tasks requiring discrimination of graphs with weak/low homophily based on their structural properties and thus high structural expressivity as found in, e.g., drug development, display a remarkable resilience to PBFA, which in some instances hardly outperformed random bit flips. Based on these observations, IBFA focuses on degrading GIN trained on tasks requiring high structural expressivity.

The discriminative power of GIN is derived from its MLPs' ability to represent injective functions on sets, as described in sec. 2. Consequently, we design our attack based on the assumption that in certain tasks where learning such a discriminative function is crucial, attacking injectivity will lead to a higher degradation than performing PBFA.

**Expressivity via injective set functions** A GNN based on message passing computing a function $F^{(k)}$ as output of the $k$th layer is maximal expressive if $F^{(k)}(u) = F^{(k)}(v) \iff c_l^{(k)}(u) = c_l^{(k)}(v)$. This is achieved when each layers' COMBINE and AGGREGATE functions are both injective, such that their combination is injective as well.

We develop the theory behind a bit flip attack exploiting this property. We formally define the concept of an unfolding tree also known as *computational tree* in the context of GNNs (D'Inverno et al., 2021; Jegelka, 2022), see fig. 1 for an example.

**Definition 1** (Unfolding Tree (D'Inverno et al., 2021)). *The* unfolding tree $T^{(k)}(v)$ *of height $k$ of a vertex $v$ is defined recursively as*

$$T^{(k)}(v) = \begin{cases} \text{TREE}(l(v)) & \text{if } k = 0, \\ \text{TREE}(l(v), T^{(k-1)}(N(v))) & \text{if } k > 0, \end{cases}$$

*where* $\text{TREE}(l(v))$ *is a tree containing a single node with label $l(v)$ and* $\text{TREE}(l(v), T^{(k-1)}(N(v)))$ *is a tree with a root node labeled $l(v)$ having the roots of the trees $T^{(k-1)}(N(v)) = \{T^{(k-1)}(w) \mid w \in N(v)\}$ as children.*

Unfolding trees are a convenient tool to study the expressivity of GNNs as they are closely related to the 1-WL colors.

**Lemma 1.** *Let $k \geq 0$ and $u$, $v$ nodes, then $c_l^{(k)}(u) = c_l^{(k)}(v) \iff T^{(k)}(u) \simeq T^{(k)}(v)$.*

Xu et al. (2019) show that GIN can distinguish nodes with different WL colors. The result is obtained by arguing that the MLP in eq. 3 is a universal function approximator (Hornik et al., 1989) allowing to learn arbitrary permutation invariant functions (Zaheer et al., 2017). This includes, in particular, injective functions. These arguments are highly theoretically and the complexity of GNNs in terms of depth, width and numerical precision required to achieve this is not well understood and subject of recent research (Aamand et al., 2022).

We investigate the functions involved in a GIN layer and how they contribute to the expressivity of the final output providing insights for the design of effective attacks. For this, it is necessary to

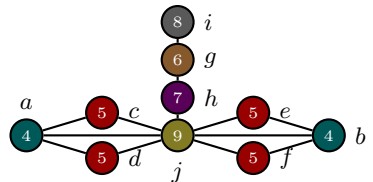

(a) Graph with final node coloring/embedding

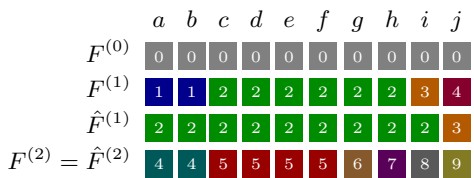

(b) Node colorings/embeddings at different layers

Figure 2: Example showing possible results of the 2-layer GNNs $F^{(2)}$ and $\hat{F}^{(2)}$ using $f^{(i)}$ and $\hat{f}^{(i)}$, respectively, for $i \in \{1, 2\}$. Nodes having the same embedding are shown in the same color and are labeled with the same integer. Although $\hat{f}^{(1)}$ is non-injective and $\hat{F}^{(1)}$ is coarser than $F^{(1)}$, we have $F^{(2)} = \hat{F}^{(2)}$. The final output corresponds to the WL coloring.

define whether we are interested in the expressivity of the general function on nodes or graphs, as in the inductive learning case, or merely the ability to distinguish the elements of a predefined subset, as in the transductive setting. First, we consider the general case, where a finite-depth GNN operates on the set of all possible finite graphs and then discuss its implication for a single concrete graph datasets. For the sake of simplicity, we limit the discussion to unlabeled graphs.

Let $f^{(i)}\colon \mathcal{M}(\mathbb{R}^{d_{i-1}}) \to \mathbb{R}^{d_i}$ be the learnable function of the $i$th layer of a GNN, where $\mathcal{M}(U)$ are all possible pairs $(A, \mathcal{A})$ with $A \in U$ and $\mathcal{A}$ a countable multisets of elements from $U$. We assume that $f^{(i)}$ is invariant regarding the order of elements in the multiset. Then the output of the network for node $v$ is obtained by the recursive function

$$F^{(k)}(v) = f^{(k)}\left(F^{(k-1)}(v), \{\!\!\{F^{(k-1)}(w) \mid w \in N(v)\}\!\!\}\right)$$

with $F^{(0)}$ uniform initial node features. Clearly, if all $f^{(i)}$ are injective, WL expressivity is reached as argued by Xu et al. (2019). The following proposition (proof in appendix) makes explicit that it suffices that all $f^{(i)}$ are injective with respect to the elements of their domain that represent (combinations of) unfolding trees of height $i - 1$.

**Proposition 1.** *Consider two arbitrary nodes $u$ and $v$ in an unlabeled graph. Let $\mathcal{J}_0$ be a uniform node feature and $\mathcal{J}_i = \{f^{(i)}(x) \mid x \in \mathcal{M}(\mathcal{J}_{i-1})\}$ the image under $f^{(i)}$ for $i > 0$. Then*

$$\forall i \le k\colon \forall x, y \in \mathcal{M}(\mathcal{J}_{i-1})\colon f^{(i)}(x) = f^{(i)}(y) \implies x = y \tag{7}$$

*implies*

$$c_l^{(k)}(u) = c_l^{(k)}(v) \iff F^{(k)}(u) = F^{(k)}(v). \tag{8}$$

This result easily extends to graphs with discrete labels and continuous attributes. The set of inputs, for which a GIN layer has to compute different outputs to achieve WL expressivity, indicates weakpoints for potential attacks. These inputs are in 1-to-1 correspondence with the unfolding trees. First, we observe that the number of unfolding trees grows quickly with increasing $i$ and increasingly discriminative functions need to be represented by the MLPs in GIN. However, Proposition 1 provides only a sufficient condition for WL expressivity. In particular, in a transductive setting when restricting to a concrete dataset, the number of different unfolding trees of each height is naturally bounded by the number of nodes. Moreover, even for a concrete dataset it is possible that the function $f^{(i)}$ applied at layer $i$ is non-injective and $F^{(i+1)}$ is still maximal expressive. Fig. 2b shows an example illustrating the situation. This motivates the need for a targeted attack on injectivity to effectively degrade expressivity, which we develop below. Further, these considerations lead to the following exemplary classification task.

**Definition 2** (WL-discriminition task). *Let $G$ be a graph with labels $l$. The* WL-discriminition task *for $k$ in $\mathbb{N}$ is to learn a function $F^{(k)}$ such that $F^{(k)}(u) = F^{(k)}(v) \iff c_l^{(k)}(u) = c_l^{(k)}(v)$ for all $u, v \in V(G)$.*

The definition, which in its above form is concerned solely with node classification based on specific structural features, can easily be extended to classifying graphs based on their structure (which we will investigate experimentally later) or structural roles (Rossi et al., 2020) refining the equitable

partition given by 1-WL's color classes. While a GNN trained on a task requiring finer partitioning than 1-WL might perform poorly, an attack targeting the GNN's ability to discriminate by a more coarse partition will at least equally damage any refinement of it. In practice, however, the number of GNN layers is kept small (e.g., $\leq 5$) to avoid overfitting (Morris et al., 2021a) or oversmoothing (Liu et al., 2020a) and full WL expressivity is not achieved. Although these shallow GNNs are limited to discriminating local structures, they still solve WL-discriminitation tasks (with low $k$) and can thus be degraded by an attack on their expressivity.

## 4  TARGETING INJECTIVITY

As revealed by our theoretical analysis in sec. 3 and illustrated by fig. 2b, it would not suffice to just consider the injectivity of a single layer's COMBINE and AGGREGATE functions for a successful attack, since maximal expressivity could be restored at deeper layers. Therefore, an attack considering the whole network is necessary. Thus, to target the injectivity required for successfully learning WL-discriminitation tasks as per def. 2 and related tasks while considering the entire model, we reformulate the target of the original PBFA from a maximization as in eq. 6 to a minimization problem

$$\min_{\{\widehat{\mathbf{W}}_{q,l}\}} \mathcal{L}\Big(\Phi(\mathbf{X}_a; \{\widehat{\mathbf{W}}_{q,l}\}_{l=1}^{L}), \ \Phi(\mathbf{X}_b; \{\widehat{\mathbf{W}}_{q,l}\}_{l=1}^{L})\Big). \tag{9}$$

That is, instead of increasing, e.g., the original classification loss of the model $\Phi$ via PBS, we use PBS to minimize the difference between the outputs of the network computed on two different inputs $\mathbf{X}_a$, and $\mathbf{X}_b$ w.r.t. the function $\mathcal{L}$ that measures the difference between the network's outputs. This approach further allows us to perform IBFA on unlabeled data.

**Choosing the loss function**  In a binary graph classification task, the network's outputs $\mathbf{y}_a = \Phi(\mathbf{X}_a; \{\widehat{\mathbf{W}}_{q,l}\}_{l=1}^{L})$ and $\mathbf{y}_b = \Phi(\mathbf{X}_b; \{\widehat{\mathbf{W}}_{q,l}\}_{l=1}^{L})$ both are $n \times 1$ vectors representing the probability mass functions (PMF) of $n$ Bernoulli distributed discrete random variables. For such distributed output vectors, any differentiable $p$-norm-based loss function would suffice to converge predictions in the sense of eq. 9 and we choose L1 for $\mathcal{L}$ for simplicity. In non-binary graph classification (i.e., multiclass-classification) or multitask binary classification, however, outputs $\mathbf{Y}_a$ and $\mathbf{Y}_b$ are not $n \times 1$ vectors but instead $n \times m$ matrices where $n$ is the number of samples and $m$ the number of classes/tasks. That is, for each of the $n$ samples, each column in $\mathbf{Y}_a$ and $\mathbf{Y}_b$ represents a PMF over $m$ classes. Thus, simply using a $p$-norm-based loss function as L1 for $\mathcal{L}$ in eq. 9 would fail to capture differences in individual class probabilities contained in $\mathbf{Y}_a$ and $\mathbf{Y}_b$ due to the reduction operation required by L1 (e.g., mean or sum over $m$). We solve this by, instead of L1, employing the discrete pointwise Kullback-Leibler-Divergence (Kullback & Leibler, 1951) (KL) as $\mathcal{L}$, i.e., the KL between the output's $n$ probability distributions of each pair of samples (data points) in $\mathbf{Y}_a$ and $\mathbf{Y}_b$, which, in the context of eq. 9, allows IBFA to find bits converging the PMF of $\mathbf{Y}_a$ best on $\mathbf{Y}_b$.

**Choosing input samples**  The proper selection of $\mathbf{X}_a$ and $\mathbf{X}_b$ is crucial as selecting inputs that have identical outputs (e.g., two batches that contain different samples of the same classes in the same order) before the attack will not yield any degradation as eq. 9 would already be optimal. To this purpose, we chose inputs $\mathbf{X}_a$ and $\mathbf{X}_b$ to be as different as possible from one another w.r.t. to the unperturbed network's outputs before the attack by iteratively solving

$$\underset{\{\mathbf{X}_a, \mathbf{X}_b\}}{\arg\max} \mathcal{L}\Big(\Phi(\mathbf{X}_a; \{\mathbf{W}_{q,l}\}_{l=1}^{L}), \ \Phi(\mathbf{X}_b; \{\mathbf{W}_{q,l}\}_{l=1}^{L})\Big). \tag{10}$$

This search mechanism can be executed before the attack itself and the found $\mathbf{X}_a$, $\mathbf{X}_b$ be reused for all iterations of the attack, a variant of IBFA to which we refer as **IBFA1**. However, after one iteration of the bit flip attack, the solution of eq. 10 might change, making it promising to recompute $\mathbf{X}_a$, $\mathbf{X}_b$ on the perturbed model before every subsequent attack iteration. We refer to the IBFA employing the latter data selection strategy as **IBFA2**. While IBFA2 may lead to slightly faster and more consistent degradation for a set amount of bit flips, its time complexity is $\Theta(kn^2)$ for $k$ attack runs and $n$ samples in the dataset, making it less suitable for large datasets. A study on the effectiveness our selection strategy when dealing with a limited subset of training data available for selection can be found in the appendix.

**Assumptions, limitations and threat model**   In line with the general trend in literature on BFAs for CNNs, we assume our target network is `INT8` quantized as exemplified in various prior works, e.g., (Yao et al., 2020; Rakin et al., 2019; 2022; Chen et al., 2021; Park et al., 2021; He et al., 2020; Li et al., 2020a; 2021; Liu et al., 2023), as such configured networks are naturally noise resistant (Rakin et al., 2022). Furthermore, we adopt the usual assumption in related work that the attacker has the capability to exactly flip the bits chosen by the bit-search algorithm through mechanisms such as RowHammer (Mutlu & Kim, 2019), NetHammer (Lipp et al., 2020) or others (Breier et al., 2018; Hou et al., 2020). Most recently, the feasability of inducing such exact bit flips via a RowHammer variant was shown by Wang et al. (2023a). We thus do not consider the detailed technical specialities of realizing the flips of the identified vulnerable bits in hardware and assume that the attacker is not subjected to budget considerations (Hector et al., 2022). Moreover, we assume some amount of training data as well as information on the network structure is available, which is in accordance with typical assumptions in related work on BFAs as found by Liu et al. (2023). This information can be acquired through methods such as side-channel attacks (Yan et al., 2020; Batina et al., 2018). Together, these typical assumptions amount to a white box threat model, whereby the attacker's goal is to crush a well-trained and deployed quantized GNN via BFA.

## 5    EXPERIMENTS

Motivated by the considerations discussed in sec. 3 and under the assumptions discussed in sec. 4, we experimentally test the hypothesis whether IBFA outperforms other BFAs on tasks requiring high structural expressivity. We assess the destructiveness of our method using real-world molecular property prediction datasets, a task common in drug development (Xiong et al., 2021; Rossi et al., 2020), as well as social network classifcation. We compare IBFA against PBFA which we consider the most relevant baseline, as most other, more specialized (e.g., targeted) BFAs designed to degrade CNNs have been derived from PBFA. We measured the degradation in the quality metrics proposed by Open Graph Benchmark (Hu et al., 2020) (OGB) or Morris et al. (2020), respectively for each of the datasets and followed the recommended variant of a 5-layers GIN with a virtual node. To ensure reproducibility, we provide details on quantized models, measured metrics, attack configuration and a code repository[1]. A detailed ablation study concerning loss function, selection strategies, layer preferences and quality degradation progression can be found in the appendix as well as experiments on GNNs less expressive than GIN and IBFA's ability to circumvent certain BFA defenses.

**Quantized models**   To evaluate our IBFA on GIN, we first obtained `INT8` quantized models by training on each dataset's training split using STE, as described in sec. 2. We used the Adam optimizer with a learning rate of $10^{-3}$ and trained the models for 30 epochs. Although more complex models and quantization techniques might achieve higher prediction quality, our focus was not on improving prediction quality beyond the state-of-the-art, but on demonstrating GIN's vulnerability to IBFA. Some of the datasets we used present highly challenging learning tasks, and our results for quantized training of GIN are comparable to those by OGB (Hu et al., 2020) for `FLOAT32` training.

**Datasets**   Six benchmark datasets are chosen (as in, e.g., (Gao et al., 2022a; Suresh et al., 2021)) from graph classification tasks from OGB based on MoleculeNet (Wu et al., 2018) for evaluation as well as `COLLAB` and `GITHUB_STARGAZERS` from TUDataset (Morris et al., 2020). The goal in each of the OGB datasets is to predict certain properties based on molecular graph structures, such that the datasets are consistent with the underlying assumptions of IBFA described in sec. 4. All OGB datasets are split using a scaffold-based strategy, which distinguishes the molecules according to their core structure and seeks to separate structurally different molecules into different subsets (Wu et al., 2018; Hu et al., 2020). `COLLAB` is derived from scientific collaboration networks, whereby every graph represents the ego-network of a scientist, and the task is to predict their area of research. `GITHUB_STARGAZERS` conatins graphs of social networks of GitHub users, and the task is to predict whether they starred popular machine learning or web development repositories. `COLLAB` and `GITHUB_STARGAZERS` are split randomly (80/10/10 for train/test/validation). More detailed dataset descriptions can be found in the appendix.

Not all targets in the OGB datasets apply to each molecule (missing targets are indicated by NaNs) and we consider only existing targets in our experiments. Area under the receiver operating charac-

---

[1]The link to our code is not included to preserve anonymity, but will be included in a final version.

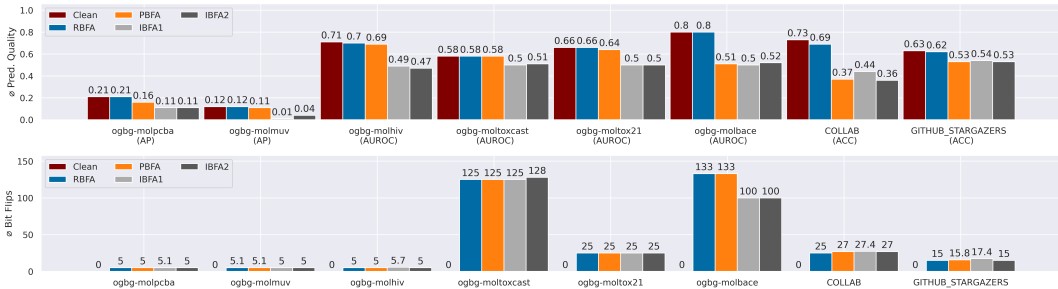

Figure 3: Pre- (clean) and post-attack test quality metrics AP, AUROC or ACC for different BFA variants on a 5-layer GIN trained on 6 `ogbg-mol` and 2 TUDataset datasets, number of bit flips, averages of 10 runs.

teristic curve (AUROC), average precision (AP) or accuracy (ACC) are used to measure the models' performance as recommended by (Hu et al., 2020) and (Morris et al., 2020), respectively.

**Attack configuration** The attacks in each of the experiments on a GIN trained on a certain dataset were executed with the number of attack runs (in the sense of attack iterations as described in sec. 2) initially set to 5 and repeated with the number of attacks incremented until the first attack type reached (nearly) random output. The other attacks in this experiment were then set to that same number of attack runs to ensure fair comparison. Note though that the evaluated attack variants PBFA, IBFA1 and IBFA2 can flip more than a single bit during one attack run (i.e. if a single bit flip did not yield improvement in the target function, combinations of 2 or more bit flips are evaluated by the algorithm), such that the final number of actual bit flips can vary across experiments even if the number of attack runs is fixed. For the single task binary classifcation datasets, `ogbg-molhiv`, `ogbg-bace` and `GITHUB_STARGAZERS`, IBFA1/2 were used with $L1$ loss, for multitask binary classifcation `ogbg-tox21`, `ogbg-toxcast`, `ogbg-molmuv`, `ogbg-pcba` and multiclass classifcation (`COLLAB`), IBFA1/2 were used with KL loss. For PBFA, binary CE (BCE) loss was used throughout the binary classification datasets and CE loss was used for `COLLAB`. Input samples for all evaluated BFA variants were taken from the training splits.

**Results** As reported in fig. 3, IBFA in both variants clearly surpasses random bit flips (**RBFA**) and PBFA in terms of test quality metric degradation for a given number of bit flips in most of the examined cases. IBFA is capable of forcing the evaluated GINs to produce (almost) random output (AUROC $\leq 0.5$, AP $\leq 0.11$ (`ogbg-molpcba`), AP $\leq 0.06$ (`ogbg-molmuv`), ACC $\leq 0.33$ (`COLLAB`) or $\leq 0.5$ (`GITHUB_STARGAZERS`)) by flipping less than 33 bits on average. IBFA2 causes slightly more destruction on `ogbg-molhiv`, `COLLAB` and `GITHUB_STARGAZERS` in terms of quality metric degradation than IBFA1 but is surpassed by or on par with IBFA1 in all other cases. IBFA2 on `ogbg-molbace` and IBFA1 on `COLLAB` were slightly weaker than PBFA. On `GITHUB_STARGAZERS`, PBFA and IBFA both degraded GIN equally. On the other hand, GINs trained on `ogbg-molmuv`, `ogbg-molhiv`, and `ogbg-moltox21` were barely affected by PBFA for the examined number of bit flips and GIN trained on `ogbg-moltoxcast` appeared to be entirely impervious to PBFA. In line with a study on `FLOAT32` GNNs (Jiao et al., 2022), our quantized GNNs also resist RBFA, ruling out the observations for PBFA and IBFA are stochastic.

## 6 CONCLUSION

The novel bitflip attack IBFA, for which we offer a sound theoretical fundament, targets specific mathematical properties of GNNs which are related to graph learning tasks requiring high structural expressivity. We illustrate IBFA's ability to exploit GIN's expressivity to render it indifferent to graph structures, compromising its predictive quality in tasks requiring structural discrimination. Specifically, we show that IBFA is clearly more destructive than the most relevant BFA variant ported from CNNs and random bit flips on eight molecular property prediction and social network classification datasets, covering binary and multiclass classification tasks. In the future, we will further investigate the robustness-efficiency relationship in GNNs as well as BFA defenses for GNNs.

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

## A    PROOFS

**Proposition 1**

*Proof.* We first prove by induction that the statement eq. 7 implies that there is a 1-to-1 correspondence between $\mathcal{J}_i$ and the isomorphism types of unfolding trees of height $i$, denoted by $\mathcal{T}_i$, for all $i \in \{0, \ldots, k\}$. In the base case $i = 0$, there is a single unfolding tree in $\mathcal{T}_0$ consisting of a single node. The uniform initialization $\mathcal{J}_0$ satisfies the requirement. Assume that $\varphi$ is a bijection between $\mathcal{J}_i$ and $\mathcal{T}_i$, then the statement eq. 7 together with the permutation-invariance guarantees that $f^{(i+1)}(A, \mathcal{A}) = f^{(i+1)}(B, \mathcal{B})$ if and only if $A = B$ and $\mathcal{A} = \mathcal{B}$. Hence,

Table 1: Preliminary case study illustrating the general vulnerability of GNNs to PBFA – pre- and post-attack mean of 10 runs of top-1 test accuracy (community) or AUROC (structure) of `INT8` quantized representative GNN architecture (GCN (Wu et al., 2022) with 3 layers, GAT (Wu et al., 2022) with 2 layers, GIN Xu et al. (2019) with 5 layers) and dataset combinations (GCN on `Cora`, GAT on `CiteSeer`, GIN on `ogbg-mol`) baseline without BFAs; after PBFA (Rakin et al., 2019) adapted to GNNs; after random bit flips (RBFA); total bit count of all model parameters (attack surface) in millions.

| | | COMMUNITY | | | | STRUCTURAL | | | | |
| --- | --- | --- | --- | --- | --- | --- | --- | --- | --- | --- |
| Attack | Dataset | Pre | Post | Flips | Bits | Dataset | Pre | Post | Flips | Bits |
| RBFA | `Cora`-GCN | 0.77 | 0.74 | 63 | 1.6M | `ogbg-molhiv`-GIN | 0.71 | 0.53 | 953 | 15.1M |
| PBFA | `Cora`-GCN | 0.77 | 0.12 | 9 | 1.6M | `ogbg-molhiv`-GIN | 0.71 | 0.50 | 953 | 15.1M |
| RBFA | `CiteSeer`-GAT | 0.58 | 0.48 | 63 | 3.8M | `ogbg-moltoxcast`-GIN | 0.58 | 0.58 | 2662 | 16.6M |
| PBFA | `CiteSeer`-GAT | 0.58 | 0.14 | 10 | 3.8M | `ogbg-moltoxcast`-GIN | 0.58 | 0.57 | 2662 | 16.6M |

$\{\!\{\varphi(a) \mid a \in \mathcal{A}\}\!\} = \{\!\{\varphi(b) \mid b \in \mathcal{B}\}\!\}$, which uniquely determines an unfolding tree in $\mathcal{T}_{i+1}$ according to def. 1. Vice versa, unfolding trees with different subtrees lead to distinguishable multisets. The result follows by Lemma 1 and the 1-to-1 correspondence shown above at layer $k$. □

## B  A MOTIVATING CASE STUDY

Our preliminary case study summarized in tab. 1 indicates a significant vulnerability of GNNs used in community-based tasks on graphs with strong homophily (Rossi et al., 2020) to malicious BFAs such as PBFA, since it suggests a quantized GNN can be degraded so severely by an extremely small number of bit flips—relative to the network's attack surface—that it produces basically random output. In our case study, a GNN's output on the community-based tasks is random if its test accuracy drops below 14.3% (= 1/7) on `Cora`'s 7-class node classification task or below 16.7% (= 1/6) on `CiteSeer`'s 6-class node classification task. Our case study shows that this is consistently the case due to the PBFA adapted from (Rakin et al., 2019) for all community-based architecture-dataset combinations examined. The number of bit flips required for completely degrading a GNN in a community-based task is remarkably small: tab. 1 shows that on average, the adapted PBFA flipped only 0.0004% of the total number of bits of the quantized GNNs' parameters. Regarding random bit flips (**RBFA**), the results of our case study are consistent with results obtained for full-precision GNNs (Jiao et al., 2022) in that they demonstrate a relatively strong resilience of GNNs against such random perturbations.

On structural tasks requiring high structural expressivity on graphs with weak/low homophily as is typical in molecular, chemical, and protein networks (Rossi et al., 2020) which are common in, e.g., drug development, PBFA is much less effective and degrades the network comparable to random bit flips. On the tasks requiring high structural expressivity in tab. 1, a GNN's output is random if its test AUROC drops to 0.5. We found that on the `ogbg-moltoxcast` dataset, PBFA could not significantly degrade the network even after 2662 flips and that on `ogbg-molhiv`, 0,0063% of the total number of bits of the quantized GNNs' parameters had to be flipped by PBFA before the GNN's output was degraded to random output, which constitutes a 15.75 times increase compared to the community-based tasks. This increased resilience of GNNs trained on tasks requiring high structural expressivity compared to community-based tasks cannot be explained entirely by the higher number of GNN parameters found in the evaluated tasks requiring high structural expressivity, which mostly stems from the MLPs employed in GIN: The increase in required flips for PBFA to entirely degrade the network on the task requiring high structural expressivity is up to 2 orders of magnitudes larger compared to the community-based task, while the increase in the attack surface is at most 1 order of magnitude larger. Based on these observations, our work focuses on such tasks requiring high structural expressivity, which are typically solved by GIN.

## C  PROGRESSIVE BIT FLIP ATTACK

The PBFA on CNN weights is an attack methodology that can crush a CNN by maliciously flipping minimal numbers of bits within its weight storage memory (i.e., DRAM). It was first introduced as an untargeted attack (Rakin et al., 2019). PBFA operates on integer quantized CNNs (as described

above) and seeks to optimize eq. 11.

$$\max_{\{\widehat{\mathbf{W}}_{q,l}\}} \quad \mathcal{L}\Big(\Phi(\mathbf{X};\ \{\widehat{\mathbf{W}}_{q,l}\}_{l=1}^{L}),\ \mathbf{t}\Big) - \mathcal{L}\Big(\Phi(\mathbf{X};\ \{\mathbf{W}_{q,l}\}_{l=1}^{L}),\ \mathbf{t}\Big)$$

$$\text{s.t.} \quad \sum_{l=1}^{L} \mathcal{D}(\widehat{\mathbf{W}}_{q,l},\ \mathbf{W}_{q,l})\ \in\ \{0, 1, \dots, N_b\} \tag{11}$$

where $\mathbf{X}$ and $\mathbf{t}$ are input batch and target vector, $\mathcal{L}$ is a loss function, $f$ is a neural network, $L$ is the number of layers and $\widehat{\mathbf{W}}_{q,l}, \mathbf{W}_{q,l}$ are the perturbed and unperturbed integer quantized weights (stored in two's complement) of layer $l$. In the original work by Rakin et al. (2019), the function $\mathcal{L}$ used is the same loss originally used during network training. $\mathcal{D}(\widehat{\mathbf{W}}_{q,l}, \mathbf{W}_{q,l})$ represents the Hamming distance between clean- and perturbed-binary weight tensor, and $N_b$ represents the maximum Hamming distance allowed through the entire CNN.

The attack is executed by flipping the bits along its gradient ascending direction w.r.t. the loss of CNN. That is, using the $N_q$-bits binary representation $\mathbf{b} = [b_{N_q-1}, \dots, b_0]$ of weights $w \in \mathbf{W}_{q,l}$, first the gradients of $\mathbf{b}$ w.r.t. to inference loss $\mathcal{L}$ are computed

$$\nabla_{\mathbf{b}}\mathcal{L}\left[\frac{\partial \mathcal{L}}{\partial b_{N_q-1}}, \dots, \frac{\partial \mathcal{L}}{\partial b_0}\right] \tag{12}$$

and then the perturbed bits are computed via $\mathbf{m} = \mathbf{b} \oplus (\text{sign}(\nabla_{\mathbf{b}}\mathcal{L})/2 + 0.5)$ and $\widehat{\mathbf{b}} = \mathbf{b} \oplus \mathbf{m}$, where $\oplus$ denotes the bitwise `xor` operator.

To improve efficiency over iterating through each bit of the entire CNN, the authors employ a method called progressive bit search (**PBS**). As noted earlier, we refer to this BFA variant employing PBS as Progressive BFA or PBFA. In PBS, at each iteration of the attack (to which we synonymously refer as *attack run*), in a first step for each layer $l \in [0, L]$, the $n_b$ most vulnerable bits in $\widehat{\mathbf{W}}_{q,l}$ are identified through gradient ranking (in-layer search). That is, regarding input batch $\mathbf{X}$ and target vector $\mathbf{t}$, inference and backpropagation are performed successively to calculate the gradients of bits w.r.t. the inference loss and the bits are ranked by the absolute values of their gradients $\partial \mathcal{L}/\partial b$. In a second step, after the most vulnerable bit per layer is identified, the gradients are ranked across all layers s.t. the most vulnerable bit in the entire CNN is found (cross-layer search) and flipped.

Should an iteration of PBS not yield an attack solution, which can be the case if no single bit flip improves the optimization goal given in eq. 11, PBS is executed again and evaluates increasing combinations of 2 or more bit flips.

## D    DATASET DESCRIPTIONS

The task associated with the `ogb-molhiv` dataset is to predict whether a certain molecule structure inhibits human immunodeficiency virus (HIV) or not. In the larger `ogb-molpcba` dataset each graph represents a molecule, where nodes are atoms, and edges are chemical bonds, and the task is to predict 128 different biological activities (inactive/active). The `ogb-moltox21` dataset contains data with qualitative toxicity measurements on 12 biological targets. The `ogbg-toxcast` dataset is another toxicity related dataset. The `obgbg-molbace` dataset is a biochemical single task binary classification (inhibition of human $\beta$-secretase 1 (BACE-1)) dataset. The `ogbg-molmuv` dataset is a subset of PubChem BioAssay commonly used for evaluation of virtual screening techniques. The `COLLAB` dataset consist of ego-networks extracted from scientific collaboration networks. In these datasets, each ego-network represents a researcher, and the objective is to forecast their specific area of research, such as high energy physics, condensed matter physics, or astrophysics. `GITHUB_STARGAZERS` contains graphs depicting GitHub users' social networks, divided based on their interactions with popular machine learning and web development repositories. (Hu et al., 2020; Gao et al., 2022a; Suresh et al., 2021; Wang et al., 2009; Morris et al., 2020). In tab. 2, an overview of the datasets' structure is provided.

## E    ABLATION STUDY

**Data selection strategies**  Fig. 4 shows that the proposed data selection strategies for IBFA1/2 provide an improvement over random data selection. Further, fig. 4 illustrates that IBFA1/2 main-

Table 2: Overview of the eight real-world benchmark datasets from OGB (Hu et al., 2020) and TUDataset (Morris et al., 2020) that are used with number of graphs, average of number of nodes and edges and recommended performance metric.

| DATASET | GRAPHS | NODES | EDGES | TASKS | METRIC | TYPE |
|---|---|---|---|---|---|---|
| ogbg-molpcba | 437929 | 26.0 | 28.1 | 128 | AP | Binary Multi-Task |
| ogbg-molmuv | 93087 | 24.2 | 26.3 | 17 | AP | Binary Multi-Task |
| ogbg-molhiv | 41127 | 25.5 | 27.5 | 1 | AUROC | Binary Single-Task |
| ogbg-moltoxcast | 8576 | 18.5 | 19.3 | 617 | AUROC | Binary Multi-Task |
| ogbg-moltox21 | 7831 | 18.6 | 19.3 | 12 | AUROC | Binary Multi-Task |
| ogbg-molbace | 1513 | 34.1 | 36.9 | 1 | AUROC | Binary Single-Task |
| GITHUB_STARGAZERS | 12725 | 391.4 | 456.9 | 1 | ACC | Binary Single-Task |
| COLLAB | 5000 | 74.5 | 4914.4 | 3 | ACC | Multi Class |

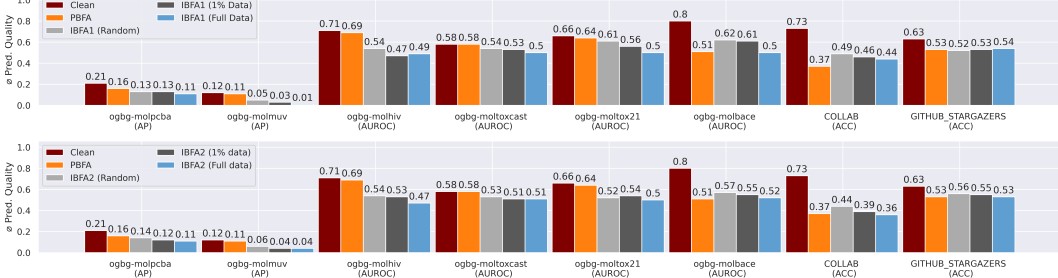

Figure 4: Pre- (clean) and post-attack test quality metrics AP, AUROC or ACC for IBFA with different data selection strategies (random, IBFA selection from 1% random subset and full data set) on a 5-layer GIN trained on 6 ogbg-mol and 2 TUDataset datasets, number of bit flips, averages of 10 runs. IBFA1 in top row, IBFA2 in bottom row.

tains a strong performance (and typically outperforms PBFA), even when utilizing a significantly constrained sample for selecting the two batches (1% subset of the dataset).

**Loss functions** Fig. 5 illustrates the results if L1 loss is used instead of KL loss in the multi-task binary classification setting. As can be seen from fig. 5, IBFA1/2 both fail to outperform PBFA on the multi-task binary classification datasets if L1 loss is used instead of KL loss, which is in line with our analytical results in sec. 4. As in fig. 4, 1% subset sampling was used for IBFA to accelerate the experiments. Experiments on obg-molbace, ogbg-molhiv and GITHUB_STARGAZERS are not included in this experiment as we already used L1 loss in our original experiments with these datasets.

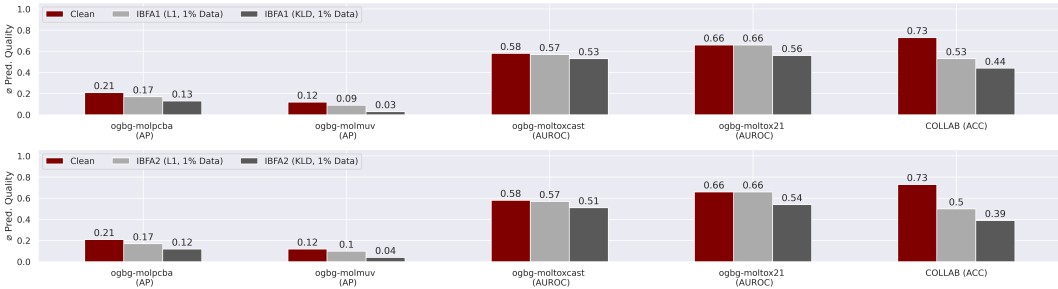

Figure 5: Pre- (clean) and post-attack test quality metrics AP, AUROC or ACC for IBFA with different loss functions (L1 vs. KL loss, IBFA selection from 1% random subset) on a 5-layer GIN trained on 4 ogbg-mol multi-task binary classification datasets and TUDataset COLLAB, number of bit flips, averages of 10 runs. IBFA1 in top row, IBFA2 in bottom row.

**Progression of degradation**   IBFA2, as shown in fig. 6, can lead to a slightly faster and more consistent degradation than IBFA1. Both IBFA variants can induce higher degradation per bit flip than PBFA. For a set amount of bit flips, this can lead to an increased overall destructiveness of the proposed method.

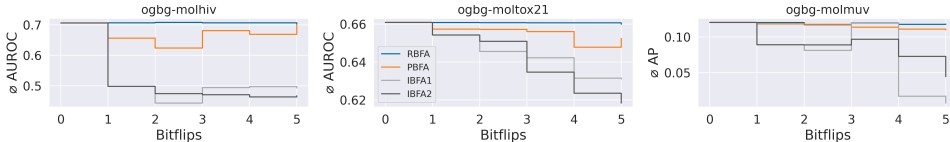

Figure 6:   Progression of quality degradation of a 5-layer GIN trained on 3 `ogbg-mol` datasets with increasing total number of bit flips induced by different BFA variants, averages of 10 runs.

**Layer preferences**   In order to investigate the attack strategies employed by the evaluated attacks PBFA, IBFA1, and IBFA2, we recorded the probabilities associated with an attack's selection of a specific component within the evaluated 5-layer GIN (refer to fig. 7). RBFA was configured to exhibit a random and uniform distribution of bit flips across the layers and is not shown in fig. 7. On the other hand, PBFA and IBFA1/2 exhibit distinct and characteristic patterns in terms of layer selection. PBFA typically confines bit flips to only 2 out of the 5 layers and, in line with Hector et al. (2022)'s findings for CNNs, displays a preference for layers near the input layer, while IBFA1/2 targets bit flips in at least 4 layers across the entire model, with the majority of flips occurring in the learnable aggregation functions of the network, namely MLP1-4. The variations in layer selection observed in IBFA1/2 support our hypotheses: a) introducing non-injectivity into a single layer alone is insufficient, necessitating an attack on the overall expressivity of GIN, and b) IBFA1/2 effectively targets the learnable neighborhood aggregation functions.

## F   OTHER GNN ARCHITECTURES

Motivated by our case study's findings as well as to obtain a first impression on IBFAs capability to generalize beyond GIN to other architectures, we repeated several experiments described in sec. 5 using Graph Convolutional Network (GCN) (Welling & Kipf, 2016) instead of GIN. In GCN, an element-wise mean pooling approach is employed for the COMBINE operation, and the steps of AGGREGATE and COMBINE are integrated in the following manner (Xu et al., 2019):

$$\mathbf{h}_v^{(k)} = \mathrm{ReLU}\left(\mathbf{W} \cdot \mathrm{MEAN}\left\{\mathbf{h}_u^{(k-1)} \mid u \in N(v) \cup \{v\}\right\}\right) \tag{13}$$

The mean aggregator used by GCN is not an injective multiset function and thus GCN's expressive power is limited (Xu et al., 2019). Fig. 8 illustrates results for a 5-layer GCN trained on 4 `ogbg-mol` datasets. As can be seen from fig. 8, IBFA1/2 outperforms or is on par with PBFA for GCN. As in fig. 4, 1% subset sampling was used for IBFA to expedite the experiments. Although not exhaustive, experiments in fig. 8 provide empirical support for our method's ability to extend beyond its initial target architecture, GIN.

## G   RELATION TO DEFENSE MECHANISMS

A cardinal aspect of BFAs is their relation with defense mechanisms (see sec. 1). It is plausible that without modifications, our attack may not successfully bypass hashing-based approaches, e.g., (Javaheripi & Koushanfar, 2021; Li et al., 2021; Liu et al., 2020b), given the absence of measures to evade fundamental assumptions in such methods (e.g., selective flipping limited to the most significant bit (MSB)). However, our approach, characterized by a unique loss function and optimization goal compared to traditional BFA attacks, will result in a distinct gradient distribution. This differentiation has the potential to challenge assumptions associated with honeypot-based defenses or gradient obfuscation techniques. To substantiate this assertion, we empirically test our hypotheses by evaluating our approach against a honeypot-based defense called Neuropots (Liu et al., 2023) and RADAR Li et al. (2021), a Run-time Adversarial Weight Attack Detection and Accuracy Recovery system. Neuropots utilizes honeypots to guide the attacker towards easily reconstructable

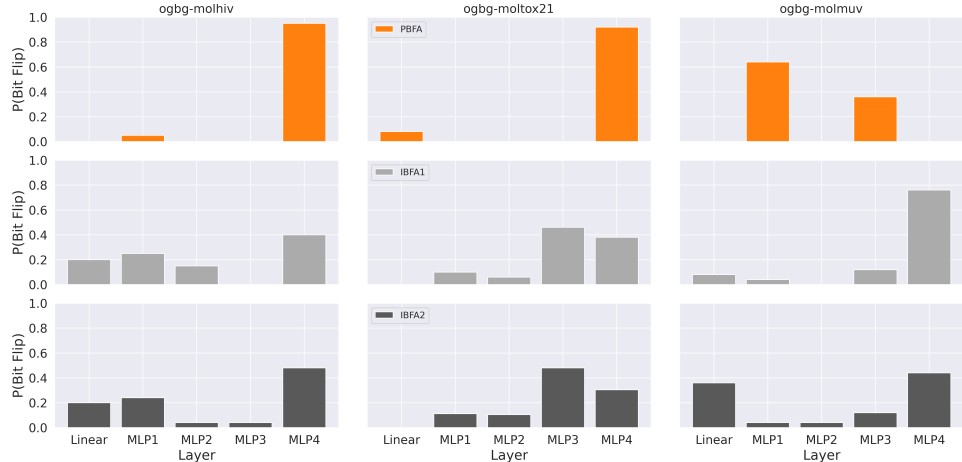

Figure 7: Probability of a component of a 5-layer GIN trained on 3 `ogbg-mol` datasets being selected for a bit flip by one of the 3 evaluated attacks, averaged over 10 runs. MLP1-4 denote the learnable neighborhood aggregation functions used in GIN, Linear denotes the linear output layer used for graph classification.

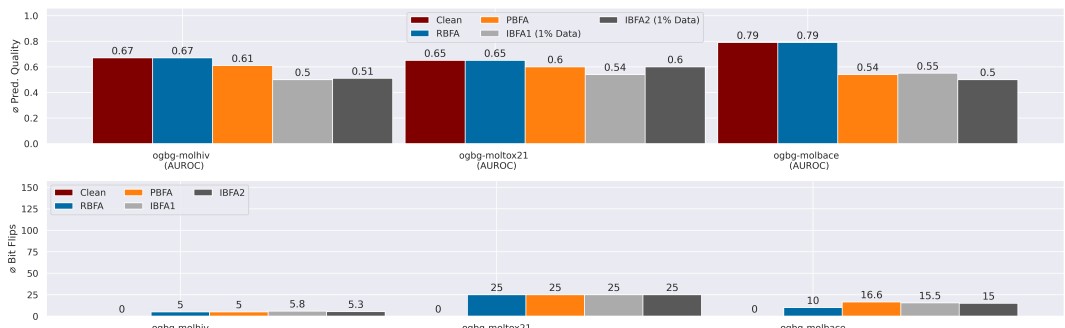

Figure 8: Pre- (clean) and post-attack test quality metrics AP or AUROC for RBFA, PBFA, IBFA1/2 (selection from 1% random subset), on a 5-layer GCN trained on 3 `ogbg-mol` datasets, averages of 10 runs.

neurons and incidentally possesses the ability to obfuscate gradients and safeguard critical neurons while RADAR groups, interleaves and hashes weights using a hash function protecting the 2 MSBs of each weight in a group. If a group is found to be compromised, the entire group is zeroed out.

**Neuropots**  Neuropots (Liu et al., 2023), which we ported from CNNs, introduces a proactive defense concept which involves the integration of a few 'honey neurons' deliberately designed as vulnerabilities within the GNN model. These vulnerabilities are strategically placed to entice potential attackers to inject faults into them, simplifying the process of fault detection and model recovery. The authors leverage Neuropots to create a defense framework that incorporates trapdoors. They devise a strategy for selecting honey neurons and propose two distinct methods for embedding trapdoors into the CNN model: a theoretically derived re-training based variant and practically more relevant heuristic variant. The retraining-based trapdoored model construction method is complex and costly, suitable only for defenders with access to ample training data. In contrast, the one-shot trapdoored model construction simplifies the process. Both approaches encompass a two-step process: initially augmenting the activation level of a specified neuron to incorporate the trapdoor, followed by fine-tuning connected weights to uphold the neuron's influence on the subsequent layer, thus reducing errors. This methodology proves effective for models utilizing full precision, albeit potentially introducing slight quantization inaccuracies in quantized models. The single-step strategy is universally applicable across all layers, inclusive of the input layer, achieved by directly modifying the activation levels of honey neurons. Moreover, given the anticipated focal point of injected

bit flips on these trapdoors, an approach utilizing checksums (specifically, the sum of the weights of the trapdoors of a layer) for detection is employed to effectively identify faults within this subset. Subsequently, the model's accuracy is reinstated through a process akin to 'refreshing' the identified faulty trapdoors. That is, compromised trapdoor neurons are replaced by their uncompromised copies, which have been kept in a safe storage.

The practically more relevant one shot encoding process used by Neuropots can be described as follows:

$$o_i^{l+1} = \sum_{j=1}^{n_l} w_{ji}^l \cdot o_j^l = w_{0i}^l \cdot o_0^l + \ldots + \left( \frac{1}{\gamma} \cdot w_{hi}^l \right) \left( \gamma \cdot o_h^l \right) \tag{14}$$

where $o_h^l$ denotes the honey neuron at layer $l$, and $w_{hi}^l$ denotes the associated honey weights. For a typical neuron, its influence on the next layer in the presence of BFAs can be formulated as $o_{l+1} = (w + \Delta w) \cdot o_l$, where $\Delta w$ denotes the weight distortion arising from bit-flips. Conversely, considering the one-shot trapdoor as an example, the impact of a 'honey neuron' on the subsequent layer can be represented as:

$$o_{l+1} = \left( \frac{1}{\gamma} \cdot w + \Delta w \right) \cdot \gamma \cdot o_l = (w + \Delta w) \cdot o_l + (\gamma - 1) \cdot \Delta w \cdot o_l \tag{15}$$

It is evident that $o_{l+1}$ experiences an increase of $(\gamma - 1) \cdot \Delta w \cdot o_l$ in comparison to a regular neuron. Furthermore, it's worth noting that attackers are more inclined to flip the MSBs of weights, resulting in a substantial perturbation $\Delta w$. Consequently, the impact of the 'honey neuron' on $o_{l+1}$ becomes more pronounced, particularly for larger values of $\gamma$. This alteration propagates and accumulates across subsequent layers, causing a substantial shift in the model's output. The honey pots that are altered during one-shot trapdoor construction are randomly chosen before model deployment and $\gamma$ is a hyperparameter provided by the user.

**RADAR** RADAR Li et al. (2021) was designed to safeguard CNN weights against PBFA. The authors systematically organize weights, distributed within a layer, into distinct groups and employ a checksum-based algorithm to generate a 2-bit signature for each group. During runtime, this 2-bit signature is computed and compared with a securely stored reference signature to identify any bit-flip attacks within a group. Upon successful detection, they zero out all weights within the affected group to mitigate the adverse impact on accuracy caused by malicious bit-flips. This proposed approach is seamlessly integrated into the inference computation stage. Commonly used detection methods like CRC or SEC-DED incur significant storage overhead and are impractical. RADAR uses simple addition-based checksum scheme and enhance its resilience against attacks by incorporating interleaving of weights and checksum on masked weights. To achieve this, RADAR calculates $M$, the sum of $G$ weights in a group, and generates a two-bit signature $Si, j = \{SA, SB\}$ from $M$ for the $i$-th layer and $j$-th group as follows

$$S_A = \left\lfloor \frac{M}{256} \right\rfloor \%2, S_B = \left\lfloor \frac{M}{128} \right\rfloor \%2 \tag{16}$$

whereby $\lfloor \ldots \rfloor$ represents the floor operation, and $\%$ signifies the remainder operation. It is important to note that in hardware, the binarization step can be achieved through straightforward bit truncation. Like the parity code, SB has the ability to identify an odd count of bit-flips on the MSBs within a group of $G$ weights. $S_B$ can't detect even bit-flips, so RADAR uses $S_A$ as a second bit. $S_A$ only detects double bit-flips if they happen in the same direction, like (0→1, 0→1) or (1→0, 1→0). Yet, flips like (0→1, 1→0) go undetected as they do not alter M's value. RADAR mitigates this vulnerability by calculating the checksum on masked and interleaved weights. RADAR applys a randomly generated secret key as a mask to a weight group, influencing whether its two's complement is considered during summation. The secret key varies in length ($N_k$ bits) from one layer to another. While a larger $N_k$ decreases the chance of correct operation sequence guessing by attackers, it raises implementation complexity. RADAR opts for $N_k$ = 16, offering ample security with $2^16$ diverse key combinations. Due to the vulnerability of addition checksum to double bit errors, an attacker can strategically target multiple bits within the same group to evade detection. To address this, RADAR calculates the checksum on a group of weights originally spaced $m$ locations apart, where $m > 1$. This technique, known as interleaving, is a recognized method used to handle burst errors in communication systems. For instance, when there are $N$ groups and each group comprises weights initially $N_W$ locations apart, the $k$-th group includes weights at positions $k + N_W l$, where $0 \leq l < N$ and $0 \leq k < NW$. By default, RADAR uses $N_W$ = G and an additional offset of 3.

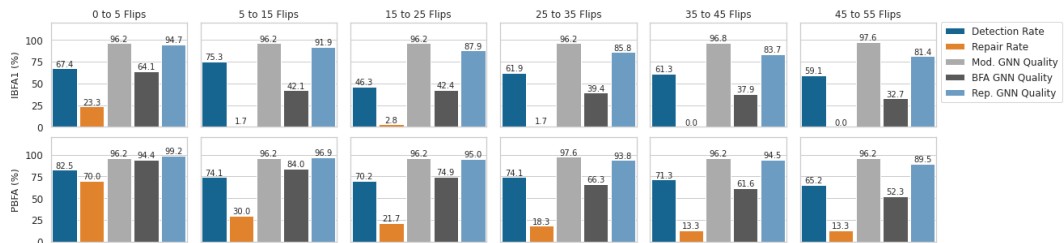

Figure 9: Neuropots statistics for reconstruction (repair), bit flip detection rates for GIN under IBFA1 (selection from 1% random subset, top) and PBFA (bottom) with varying amount of bit flips, aggregated means across datasets, 10 runs per dataset per bar chart (460 runs in total). Rescaling parameter set to $\gamma = 2.0$ (as proposed in the original paper) and honeypot percentage set to $5\%$. GNN quality refers to pre- and post-attack quality metrics relative to the clean (unperturbed) model.

**Results** For this series of experiments, we used an identical experimental setup as previously described in sec. 5, except that we gradually increased the number of attack runs (and therefore bit flips) to find out up to which point Neuropots and RADAR have a good chance of reconstructing a model attacked by PBFA/IBFA1. We omitted IBFA2 experiments to reduce runtime as it bears great similarity to IBFA1.

As can be seen from fig. 9, which displays aggregated statistics over all evaluated datasets, IBFA1 significantly reduces bit flip detection and GNN reconstruction (i.e., the repaired GNN is identical to the GNN before being subjected to BFA) rates of Neuropots compared to PBFA. The best observed reconstruction rate of Neuropots for IBFA1 was 23% wherby models attacked by PBFA could be fully reconstructed via Neuropots in up to 70% of the runs of 0-5 bit flips experiment. Furthermore, despite Neuropots capability to actively steer a BFA attacker away from the most vulnerable neurons and towards randomly selected trapdoor neurons, IBFA1 is capable of selecting weights in neurons that degrade the model significantly more than PBFA. Fig. 10 displays the same statistics as fig. 9

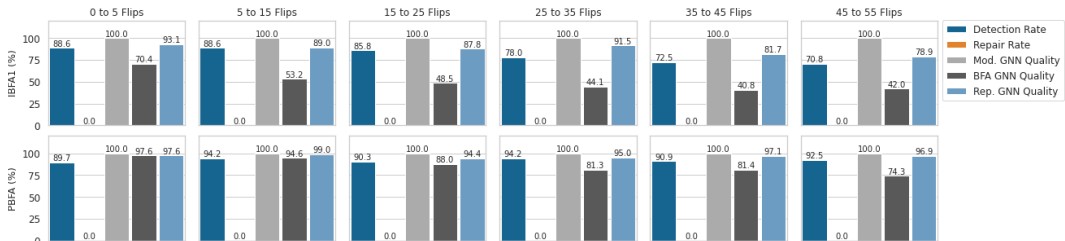

Figure 10: RADAR statistics for reconstruction (repair), bit flip detection rates for GIN under IBFA1 (selection from 1% random subset, top) and PBFA (bottom) with varying amount of bitflips, aggregated means across datasets, 10 runs per dataset per bar chart (460 runs in total). Group size $G = 16$, interleaving offset $m = 3$. GNN quality refers to pre- and post-attack quality metrics relative to the clean (unperturbed) model.

for RADAR. While RADAR, as a hashing-based approach, cannot easily be bypassed by IBFA without stealth modifications (such as, e.g., exploiting the fact that RADAR limits protection to the 2 MSBs of each weight) as easily as Neuropots, IBFA1 detection rates are still significantly lower than for PBFA, which, given RADARs operating mechanism, implies that IBFA1 more often flips bits other than the MSB 2 bits compared to PBFA. Additionally, for a set amount of bit flips, RADAR's capability to repair the attacked GNN's prediction quality after IBFA1 is notably limited compared to PBFA. This indicates an inherent ability of IBFA1 to circumvent hashing-based defense strategies and future evaluation of stealthy IBFA variants might be worthwhile.

