# OpenReview forum: "Attacking Graph Neural Networks with Bit Flips: Weisfeiler and Lehman Go Indifferent"
_ICLR.cc/2024/Conference — Submitted to ICLR 2024_

### Official Review · Reviewer_Ve7w · 2023-10-27

**Soundness:** 3 good
**Presentation:** 2 fair
**Contribution:** 3 good
**Rating:** 5
**Confidence:** 3

**Summary:**

In this paper, the authors investigated the susceptibility of GNNs to PBFA. They introduced the Injectivity Bit Flip Attack (IBFA), a novel attack targeting the discriminative power of neighborhood aggregation functions used by GNNs. IBFA differs from existing BFA variants for CNNs by its bit-search algorithm’s optimization goal as well as input data selection strategy and is distinct from graph poisoning and evasion attacks as input data is left.

**Strengths:**

1. The authors provided a strong theoretical fundament for IBFA which is confirmed by extensive experimental evidence of its effectiveness on real-world datasets under the assumptions common in established BFA research.
2. It is novel to design BFA of GNNs targeting the discriminative power of neighborhood aggregation functions used by GNNs.

**Weaknesses:**

1. The paper spends too many pages discussing preliminaries in the main paper. However, the IBFA is not well introduced.
2. The author only demonstrates the application of IBFA on GIN and GCN. What about other popular graph neural networks such as GAT, GraphSAGE, and Graphormer?
3. What is the computation cost of IBFA?
4. Is IBFA practical in real-world situations?
5. What are the potential defense methods? Can IBFA be easily detected?

**Questions:**

Please see the weakness.

---

### Official Review · Reviewer_BAQd · 2023-11-01

**Soundness:** 2 fair
**Presentation:** 2 fair
**Contribution:** 2 fair
**Rating:** 3
**Confidence:** 3

**Summary:**

This paper concerns security aspects of GNNs. In particular, whereas previous works primarily focused on attack related to the graph and feature information, the current work focusses on attacks on the weights and biases of the GNNs. Inspired by work on (progressive) Bip Flip Attacks (BFA) on CNNs, the impact of such attacks on the accuracy of GNNs is explored. The work on progressive bit search by Rakin et al (2019) serves as additional inspiration. Furthermore, the authors leverage the characterisation of the expressive power of GNNs in terms of color refinements and tree unravellings.

At the core of the proposed method lies the observation that harmful bit flips should destroy injective at all layers of the GNN. This results in so called Injective Bit Flip Attacks (IBFA) for short.

**Strengths:**

1. The related work is well described.
2. The idea to use bit flips to destroy injective is nice.
3. The proposed methods is experimentally verified.

**Weaknesses:**

1. The need for investigating security aspects of GNNs needs better motivation. Is this really problem originated from practical need?
2. The theoretical framework section is trivial. It is unclear what is the new insight here.
3. The presentation is confusing. More details should be given as to how injectivity as attacked.
4. The iterative method in section 4 should be explained better.

**Questions:**

**Q0.** Is the need to protect GNNs from attacks a practically relevant problem? Please motivate by use cases.

**Q1.**  Please explain more clearly how equation 9 is used to attack injectivity.

**Q2.** What do you mean by "iteratively solving ... eq 10". What is the iterative aspect?

**Q3.** Please clarify Figure 2b. It is unclear what is going on.

---

### Official Review · Reviewer_DkBB · 2023-11-02

**Soundness:** 2 fair
**Presentation:** 4 excellent
**Contribution:** 2 fair
**Rating:** 3
**Confidence:** 4

**Summary:**

This study proposed a novel bit-flip attack (BFA) approach tailored to GNNs. The attack leverages a distinctive loss method that capitalizes on the need for injective MLP in GNNs, enabling them to match the expressiveness of the 1-WL test. Compared to other BFAs, this attack demonstrates superior performance in the number of bits flipped and the model's overall efficacy.

**Strengths:**

1. The idea of minimizing the difference between the embeddings of two different (non-isomorphic graphs) to reduce the expressivity of GNNs is exciting and novel.
2. The paper is clear and easy to follow.

**Weaknesses:**

1. I did not find formal proof in the work, theoretical or empirical, for the claim that the suggested BFA causes the attacked GNNs to lose the expressivity of the Weisfeiler-Lehman test.

2. Other GNN models with more sophisticated message-passing techniques and better expressivity, such as [1],[2], and [3], should also be tested in addition to GIN.

3. The suggested BFA should take into account that even simple methods such as GIN may use pre-coloring (or preprocessed node features) such as [4] and [5] to improve their performance and expressiveness beyond the Weisfeiler-Lehman test. These preprocessing techniques might defend GIN and other methods from these kinds of attacks, at least in terms of expressivity.

4.  Additional attacks other than PBFA should be compared to IBFA.


[1] "Weisfeiler and leman go neural: Higher-order graph neural networks." by Morris et al.

[2] " Weisfeiler and lehman go cellular: Cw networks.", by Bodnar et al.

[3] "Weisfeiler and Lehman go topological: Message passing simplicial networks." by Bodnar et al.

[4] "Improving graph neural network expressivity via subgraph isomorphism counting." by Bouritsas et al.

[5] "Weisfeiler and Leman Go Infinite: Spectral and Combinatorial Pre-Colorings", by Feldman et al.

**Questions:**

Please see the section above.

I am expecting to see authors' response to the concerns raised in "Weaknesses"

---

### Official Review · Reviewer_5CWb · 2023-11-03

**Soundness:** 2 fair
**Presentation:** 2 fair
**Contribution:** 2 fair
**Rating:** 5
**Confidence:** 3

**Summary:**

This paper proposes the "Injectivity Bit Flip Attack" (IBFA), a bit-flip attack tailored for graph neural networks. This attack targets learnable neighborhood aggregation functions, degrading their ability to distinguish graph structures and undermining the expressiveness of the Weisfeiler-Lehman test. The paper suggests that exploiting mathematical properties specific to certain graph neural network architectures significantly increases their vulnerability to bit flip attacks. Injectivity Bit Flip Attacks (IBFA) can reduce the output of Graph Isomorphism Networks trained on various graph prediction datasets to random values with just a small fraction of bit flips, demonstrating its effectiveness.

**Strengths:**

+ The paper explores a niche research direction that has received limited attention.
+ The attack proposed in the paper demonstrates some better empirical results compared to other considered baseline methods.

**Weaknesses:**

- The presentation of the paper is challenging to follow, and the flow of the content needs significant improvement. This may hinder the reader's ability to understand and appreciate the research.

- The theoretical foundations of the paper are not explicitly strong, and it remains unclear how the proposed approach contributes to a stronger attack. The lack of a well-defined algorithm or framework to connect all the ideas together leaves gaps in the paper's coherence.

Overall, the current version of the paper fails to clearly communicate its contribution, making it difficult for readers to grasp the significance of the research. A more structured and comprehensive explanation of the methodology and its implications is needed to address this issue.

**Questions:**

"Our findings suggest that exploiting mathematical properties specific to certain graph neural network architectures can significantly increase their vulnerability to bit flip attacks."  Could you please specify which section provides a detailed explanation of this point? Additionally, I'm curious about the potential computational cost associated with exploiting these mathematical properties, as it may require domain-specific expertise and could be resource-intensive.

Is there any proof available for Lemma 1? I couldn't find it in the Appendix.

Could you clarify the primary contributions of this paper? The experimental results appear to lack significant impact in some scenarios, and the theoretical contributions are not clearly explained in terms of enhancing the attack's effectiveness and strength. The paper also suggests that the idea is not entirely novel.

---

### Official Review · Reviewer_QF5L · 2023-11-09

**Soundness:** 3 good
**Presentation:** 3 good
**Contribution:** 3 good
**Rating:** 5
**Confidence:** 2

**Summary:**

The authors first analyzed the models' representation ability from the WL theory and then they proposed a new effective bit flip attack on graph isomorphism networks. By degrading their ability to distinguish graph structures and losing the expressivity of WL test. Their blt flip attack can degrade GIN's performance on various graph property prediction datasets.

**Strengths:**

I summarize their strengths as follows:
1. The author's writing is easy to follow.
2. Their theory and empirical results are well supported.
3. They are the first ones to consider the bit flip attack in graph models. And their empirical results also demonstrate the effectiveness of their attacks.

**Weaknesses:**

I think the weakness is their analysis and the scenario of such an attack. I leave them in the questions.

**Questions:**

However, I also have some questions:
1. I don't know whether attacks on quantized GNNs are worth exploring. Because unlike vanilla CNNs or transformers, GNN models especially GINs are usually small. And users usually train a new model on their specific tasks. So I doubt whether such an attack's influence and whether such an attack is worth exploring.
2. I think your analysis can also work for graph injection attacks or modification attacks. Then why these models can work well by only attacking the CE loss? Could you please use your theory to discuss why graph injection attack can work well and traditional bit flip attack cannot?

---

### Meta-Review · Area_Chair_yGVf · 2023-12-08

**Metareview:**

The paper introduces a unique approach to attacking GNNs, but it faces several key concerns. Reviewers questioned the practical relevance of the proposed attack, especially in real-world scenarios, and noted the need for clearer presentation and better theoretical foundations. Additionally, there were concerns about the limited testing on GNN models and the lack of clarity on computational costs and potential defense methods. The paper cannot meet the acceptance threshold due to these issues.

**Justification For Why Not Higher Score:**

as said in meta review

**Justification For Why Not Lower Score:**

NA

---

### Decision · Program_Chairs · 2024-01-16

Reject